# Challenging the "old boys club" in academia: Gender and geographic representation in editorial boards of journals publishing in environmental sciences and public health

Sara Dada[1], Kim Robin van Daalen[2]*, Alanna Barrios-Ruiz[3], Kai-Ti Wu[4,5], Aidan Desjardins[6], Mayte Bryce-Alberti[7], Alejandra Castro-Varela[3], Parnian Khorsand[8], Ander Santamarta Zamorano[9], Laura Jung[10], Grace Malolos[11], Jiaqi Li[12], Dominique Vervoort[13], Nikita Charles Hamilton[14,15], Poorvaprabha Patil[16], Omnia El Omrani[17], Marie-Claire Wangari[18], Telma Sibanda[19], Conor Buggy[20], Ebele R. I. Mogo[21]

1 UCD Centre for Interdisciplinary Research, Education and Innovation in Health Systems, School of Nursing, Midwifery and Health Systems, University College Dublin, Dublin, Ireland, 2 Department of Public Health and Primary Care, Cardiovascular Epidemiology Unit, Cambridge University, Cambridge, United Kingdom, 3 Escuela de Medicina y Ciencias de la Salud, Tecnológico de Monterrey, Monterrey, México, 4 Leibniz Institute for Freshwater Ecology and Inland Fisheries, Berlin, Germany, 5 Faculty of Mathematics and Natural Science, Department of Geography, Humboldt University of Berlin, Berlin, Germany, 6 Department of Microbiology, School of Genetics and Microbiology, Moyne Institute of Preventive Medicine, Trinity College Dublin, Dublin, Ireland, 7 Facultad de Medicina, Universidad Peruana Cayetano Heredia, Lima, Perú, 8 Women in Global Health, Washington, District of Columbia, United States of America, 9 School of Nursing and Midwifery, Trinity College Dublin, Dublin, Ireland, 10 Leipzig University, Medical Faculty, Leipzig, Germany, 11 College of Medicine, University of the Philippines Manila, Manila, Philippines, 12 University of Cambridge, School of Clinical Medicine, Addenbrooke's Hospital, Cambridge, United Kingdom, 13 Institute of Health Policy, Management and Evaluation, University of Toronto, Toronto, Ontario, Canada, 14 NCH Strategy Group, Nassau, The Bahamas, 15 The Department of Environmental Planning and Protection (DEPP), Nassau, Bahamas, 16 Kasturba Medical College, Manipal Academy of Higher Education, Manipal, Karnataka, India, 17 Faculty of Medicine, Ain Shams University, Cairo, Egypt, 18 Nazareth Hospital, Nairobi, Kenya, 19 Zimbabwe Red Cross Society, Harare, Gokwe South and North, Zimbabwe, 20 School of Public Health, Physiotherapy and Sports Science, University College Dublin, Dublin, Ireland, 21 MRC Epidemiology Unit, University of Cambridge, Cambridge, United Kingdom

☯ These authors contributed equally to this work.
* k.r.vandaalen@gmail.com

**Data Availability Statement:** The data that support the findings of this study are openly available at:

## Abstract

In light of global environmental crises and the need for sustainable development, the fields of public health and environmental sciences have become increasingly interrelated. Both fields require interdisciplinary thinking and global solutions, which is largely directed by scientific progress documented in peer-reviewed journals. Journal editors play a critical role in coordinating and shaping what is accepted as scientific knowledge. Previous research has demonstrated a lack of diversity in the gender and geographic representation of editors across scientific disciplines. This study aimed to explore the diversity of journal editorial boards publishing in environmental science and public health. The Clarivate Journal Citation Reports database was used to identify journals classified as Public, Environmental, and Occupational (PEO) Health, Environmental Studies, or Environmental Sciences. Current EB members were identified from each journal's publicly available website between 1 March

https://github.com/KimvanDaalen/EBrepresentation and presented in the Supplement Materials. These data were derived from the publicly available webpages of included journals in this study.

**Funding:** The authors received no specific funding for this work. KRvD receives funding by the Gates Cambridge Scholarship (OPP1144) for her PhD research, and received funding for publication from the Bill and Melinda Gates Foundation.

**Competing interests:** The authors have declared that no competing interests exist.

and 31 May 2021. Individuals' names, editorial board roles, institutional affiliations, geographic locations (city, country), and inferred gender were collected. Binomial 95% confidence intervals were calculated for the proportions of interest. Pearson correlations with false discovery rate adjustment were used to assess the correlation between journal-based indicators and editorial board characteristics. Linear regression and logistic regression models were fitted to further assess the relationship between gender presence, low- and middle-income country (LMIC) presence and several journal and editor-based indicators. After identifying 628 unique journals and excluding discontinued or unavailable journals, 615 journal editorial boards were included. In-depth analysis was conducted on 591 journals with complete gender and geographic data for their 27,772 editors. Overall, the majority of editors were men (65.9%), followed by women (32.9%) and non-binary/other gender minorities (0.05%). 75.5% journal editorial boards (n = 446) were composed of a majority of men (>55% men), whilst only 13.2% (n = 78) demonstrated gender parity (between 45–55% women/gender minorities). Journals categorized as PEO Health had the most gender diversity. Furthermore, 84% of editors (n = 23,280) were based in high-income countries and only 2.5% of journals (n = 15) demonstrated economic parity in their editorial boards (between 45–55% editors from LMICs). Geographically, the majority of editors' institutions were based in the United Nations (UN) Western Europe and Other region (76.9%), with 35.2% of editors (n = 9,761) coming solely from the United States and 8.6% (n = 2,373) solely from the United Kingdom. None of the editors-in-chief and only 27 editors in total were women based in low-income countries. Through the examination of journal editorial boards, this study exposes the glaring lack of diversity in editorial boards in environmental science and public health, explores the power dynamics affecting the creation and dissemination of knowledge, and proposes concrete actions to remedy these structural inequities in order to inform more equitable, just and impactful knowledge creation.

## Introduction

For the last several decades, the relationship between the environment and public health has been examined across disciplines [1,2]. This link has become more apparent, particularly the need for sustainable development and advancement of planetary health as we face the interlinked global environmental crises of climate change and biodiversity loss, exemplified by the recent COP26 [3–5]. The global repercussions of these challenges, requires interdisciplinary and multilateral responses that connect the global to the local. It is important to consider not only political implications, but also pedagogies around these fields to promote sustainability and equity [6,7]. This begins with knowledge creation and dissemination, exemplified by what is published as scientific literature in peer-reviewed, academic journals.

While women outnumber men in academia, the vast majority of academic leadership positions are held by men [8,9]. This pattern is further evidenced by the composition of journal editorial boards. Over the years, several published studies have highlighted the under-representation of women on editorial boards of major academic journals [10–20]. For example, a 2011 analysis of the editorial boards of 60 top-ranked medical journals found that 17.5% of the 4,112 editorial board members and only 10 of the 63 editors-in-chief were women [14]. A decade later, women are still under-represented and make up about one in five editors-in-chief across over 400 medical journals [19]. This representation gap is even more glaring for

certain specialties: a similar analysis of 42 American surgical journals in 2019 revealed only two editors-in-chief and less than 15% of the editorial boards were women [17]. While this pattern has been seen across various disciplines such as mathematics [20], cardiology [16], global health [11], infectious diseases and microbiology [15], and environmental biology [21], it is important to also note that few studies include individuals identifying outside the gender binary, such as gender non-confirming or transgender* individuals.

In addition to exposing a lack of gender diversity, studies examining geographic representation and socio-economic diversity of academic journal editorial boards have found similar results [11,18,22]. Illustratively, an analysis of 24 ecology and environmental biology journals over three decades (1985–2014) found that almost 70% of all editors in that period were based in the United Kingdom or the United States, whilst editors based outside high-income countries (HICs) were extremely rare [18]. Furthermore, in the top ten international psychiatry journals in 2016, only 21 (3.5%) of 607 editorial board members were from low- or middle-income countries (LMICs) [23]. Notably, 73% of the examined 303 editors of 27 global health journals were based in HICs; 64% were in Europe, Central Asia, and North America, exhibiting a definition of 'global' that is found lacking [11]. When considering intersectionality of gender and geography, only 4% of editors with leadership roles from 12 major global health journals were women from LMICs [22]. These statistics are a shocking revelation of academic power imbalances, considering that over 80% of the world's population lives in LMICs [24].

A journal editor's role is to manage and coordinate manuscript submissions, implying an ability to influence what is and is not published. Editorial boards and their leadership shape the direction of future progress and research [18,25–27]. Imbalanced representation on such boards may therefore both reflect and result in biases and systemic disparities as well as power asymmetries in the production of scientific knowledge [18,23,28]. Diverse editorial boards offer a range of benefits, including an increased openness to innovative approaches, increasing manuscript submissions, enhancing the impact and profile of a journal and its articles, ensuring the accuracy, representation and interpretation of data from the countries studied, and building the capacity of academic societies and scholars across socio-economic strata [18,23]. Calls to decolonize academia and its sub-specialties (e.g. global health) are growing, highlighting both individuals' and organizations' roles in addressing and responding to power asymmetries in academia [26,29,30]. This movement calls attention to how a "White, Western" lens shapes the way knowledge is produced and learned and how structural changes are needed to incorporate the perspectives, knowledge and lived experiences of gender and ethnic minorities in curricula, teaching, and learning [31,32] Addressing the ways journal editorial boards are formed and operate represent an integral component in amplifying progress in this movement.

The gender and geographic representation of editorial boards has been minimally considered in the full scope of public, environmental, and occupational (PEO) health and in the environmental science disciplines–including their intersecting research disciplines such as planetary health and the impacts of climate change on health. These research areas are growing as global environmental crises unfold and are becoming a central part of finding solutions for sustainable development. The work published today contributes to scientific progress and future policy-making, highlighting the need to make diverse voices heard. Therefore, the objective of this study is to examine the inferred gender and geographic/economic representation of journal editorial boards that publish in 1) PEO health journals, and 2) environmental science/studies. This analysis considers the broader spectrum of environment and public health journals, in comparison to research focused on specific disciplines. In doing so, this study contributes to the exploration of power dynamics in the creation and dissemination of knowledge.

## Methods

### Data collection

The Clarivate Journal Citation Reports (JCR) was used to identify all journals classified in 1) either of the PEO Health categories, 2) the Environmental Studies category, or 3) the Environmental Sciences category (https://jcr.clarivate.com/). These categories were used to capture journals that publish research in the fields of health and environment. Journals with no publicly available information about their editorial boards and exclusively university-affiliated student-run journals were excluded.

Current editorial board members were identified from each journal's publicly available website between 1 March and 31 May 2021. We included i) current journal editors-in-chief, ii) associate, deputy, section, or senior editors, and iii) editorial board members or editorial committee members (including advisory and review boards). We excluded editors that were honorably retired (e.g., editor emeritus), managing editors, and blog, social media, or news editors. As journals often have different titles for positions with similar responsibilities, we assigned editorial board members to one of five categories depending on their primary responsibilities (Table 1): Editor-in-Chief (EiC), Editorial Leadership (EL), Editorial Board (EB), Advisory Board (AB), Early Career (EC). To distinguish these terms, Editorial Team (ET) is used to describe journals' overall editorial groups (EB and AB). For journals with > 1000 members on their EB, we only extracted information on their EiC and EL *(Frontiers in Environmental Science*, *Frontiers in Public Health*, *International Journal of Environmental Research & Public Health*, and *Sustainability)*. This decision was made on the basis that these journals often included peer-reviewers in their EB as 'review-editors.'

**Table 1. Editorial board members' roles and categories.**

| Role category | Variations of title(s)/role(s) |
|---|---|
| **Editor-in-Chief (EiC)** | • Editor-in-chief, co-editors-in-chief, chief editor |
| **Editorial Leadership (EL)** | • Editor-in-chief, co-editors-in-chief, chief editor<br>• Deputy editor in chief, deputy editor, senior deputy editor, associate editor-in-chief, chair, senior editor, senior associate editor, section editor, associate editors, special issue editors, chairs, editorial director, executive editor, commissioning editor, associated editor |
| **Editorial Board (EB)** | • Editor-in-chief, co-editors-in-chief, chief editor<br>• Deputy editor in chief, deputy editor, senior deputy editor, associate editor-in-chief, chair, senior editor, senior associate editor, section editor, associate editors, special issue editors, chairs, editorial director, executive editor, commissioning editor, associated editor<br>• Editorial board member, editor, editorial member, editorial officer, consulting editors, statistics/reviews editors |
| **Advisory Board (AB)** | • Editorial advisory board, editorial advisory panel, advisory panel, international advisory board, national advisory board, editorial advisor, consulting board |
| **Youth/Early Career (EC)** | • Young editorial member, early-career mentored editorial board |
| **Editorial Team (ET)** | • Editor-in-chief, co-editors-in-chief, chief editor<br>• Deputy editor in chief, deputy editor, senior deputy editor, associate editor-in-chief, chair, senior editor, senior associate editor, section editor, associate editors, special issue editors, chairs, editorial director, executive editor, commissioning editor, associated editor<br>• Editorial board member, editor, editorial member, editorial officer, consulting editors, statistics/reviews editors<br>• Editorial advisory board, editorial advisory panel, advisory panel, international advisory board, national advisory board, editorial advisor, consulting board<br>• Young editorial member, early-career mentored editorial board |
| *Excluded Categories* | • Assistant editors, book review editors, language editors, managing editors, non-editorial executive teams, non-editorial director, technical editors |

We collected the following information on each editorial team member: full name, role on editorial team, institutional affiliation, geographic location (city and country), and inferred gender. In cases where an author's institutional affiliation was not stated on the journal's website, the affiliation stated on their latest scientific publication or online biography was utilized. Where more than one institution was listed, we used the initial or primary affiliation provided. We assigned countries in which the editors' institutions were based to their corresponding United Nations (UN) region group (Africa, Asia and Pacific, Eastern Europe, Latin America and Caribbean, Western Europe & Others), World Bank income group (high-income, upper-middle income, lower-middle income, low-income), and Gender Inequality Index (GII). Higher GII values correspond to increased disparities between women and men in these areas [33]. Whilst the country in which an editor is based is often conflated with an editor's nationality, these are not interchangeable. Our analysis specifically explores the country, UN region, and World Bank income group of the countries in which the editor's institution is based without using this as a proxy for nationality.

For each journal we collected the following information from the Scimago Journal & Country Rank (SJR) based on 2020 data (https://www.scimagojr.com/): H-index, journal total cites, publisher, country of journal, and time coverage. For two journals (Journal of Geophysical Research-Biogeosciences and Sex Education-Sexuality Society and Learning) we used the SJR data for their parent journals (Journal of Geophysical Research and Sex Education, respectively). We used the JCR journal indices to assign journals to one of the following categories: those indexed as Public, Occupational and Environmental health, those indexed as Environmental Sciences, and those indexed as both. The 2020 journal impact factor (IF) was obtained from JCR. Countries of journal publication were assigned their UN region, World Bank income group, and GII.

## Inferring likely gender

The likely gender of editorial team members was inferred manually for 92.4% of included editors (n = 26,453) based on gendered prefixes (e.g., Ms., Mrs. Miss, Mr., Mx.) or gendered pronouns (e.g., she/he/they) on journal websites, online biographies and/or social media profiles. In cases where gender could not be inferred, a gender-to-name algorithm (https://genderize.io/) was used (7.6%, n = 2,170) to infer likely gender based on historical databases combining first names and country. This tool has been checked for robustness and applied in multiple previous studies [34–36]. The algorithm's inferred gender was accepted only when the probabilistic certainty score was ≥0.90. If the probability score was below 0.90, the inferred gender was considered "unknown;" this applied to 1.7% (n = 485) of the individuals. Due to the inability of such algorithms to identify people outside the gender binary and their reduced quality for inferring gender for non-Western names, this option functioned as a last resort.

## Statistical analysis

Descriptive statistics were calculated at the editor-level and journal-level. For analyses at the journal level, journals with >10% missing ("unknown") for inferred genders were excluded (n = 24). Inferred gender (women and gender minorities vs. men) and country income group (LMICs vs. HICs) were both dichotomized as well as categorized into three categories. Gender composition was categorized into majority women and gender minorities, gender parity (45–55%) and majority men. Income group was categorized into majority LMICs, economic parity (45–55%), majority HICs. Mean GII indexes, percentage of women and gender minorities, and the percentage of LMIC-based editors were also calculated by editor role. Missing values were excluded from the analyses.

Binomial 95% confidence intervals (CIs) were calculated for the proportions of interest. Pearson correlations with False Discovery Rate (FDR) adjustment for multiple testing were used to assess the correlation between journal-based indicators, women and gender minority presence, LMIC presence and GII indexes by EiC, EL, EB, and AB. Linear regression models and logistic regression models were fitted to further assess the relationship between gender presence, LMIC presence, and several journal and editor-based indicators. Sensitivity-analyses were used to determine how different values of the independent variables affected the dependent variable. *P*-values below 0.05 were considered statistically significant.

All statistical analyses and data visualization was conducted in STATA version 16 and R version 4.0.5 (R Foundation, Vienna, Austria, www.r-project.org). For data visualization the tidyverse, dplyr, ggplot and ggcorplot packages were used. The data from this study is available at: https://github.com/KimvanDaalen/EBrepresentation and presented in the Supporting Information.

### Ethical considerations

The research team consulted with academics experienced in human research ethics and this study was not considered to have any ethical issues. All data used for this study was not restricted nor sensitive, nor did it require permission to access or collate. Data was publicly available and accessible, eliminating the need for additional ethical approval.

### Research team

The research team was composed of an internationally diverse group of young researchers from a wide variety of cultural backgrounds, with proficiency in a number of different languages (Arabic, Basque, Dutch, English, Farsi, Filipino, French German, Hindi, Japanese, Kikuyu, Kiswahili, Mandarin, Marathi, Portuguese, Shona, Spanish, Telegu). Additionally, multiple members of the research team live or work in countries that are different from the country of their nationality. This wide geographic base and language knowledge allowed the team to include non-English/non-Western sources and perspectives.

## Results

After exporting the results from the four JCR categories, we identified a total of 628 unique journals. Thirteen journals were excluded that either lacked publicly available information, had been discontinued/merged/renamed, or were exclusively student publications (**S1 Table**), resulting in 615 journals with 28,832 editors. For additional levels of analysis, we further excluded journals with >10% unknowns for inferred gender (n = 24, **S2 Table**) resulting in a total of 591 included journals with 27,722 editors (**Fig 1**).

**Table 2** exhibits overall summary characteristics of all the collected data. Editorial teams ranged in size from 5–736 individuals and their institutions were based in 150 countries. Journals were published by 164 unique publishers and based in 35 countries. While the majority of journals were composed of EB and/or ABs, thirteen journals also included "early career/young professionals" in the editorial/advisory board.

### Descriptive statistics

Across all journals, 25% of EiC, 35% of EL, 33.3% of EB, 30.5% of AB, 51.8% of EC and 33% of all editorial teams were inferred as women and gender minorities (WGM). Seventy-eight journals (13.2%) displayed gender parity (GP), 67 journals (11.3%) had a majority of WGM, and 446 journals (75.5%) had a majority of men. Journals classified as PEO Health had the greatest

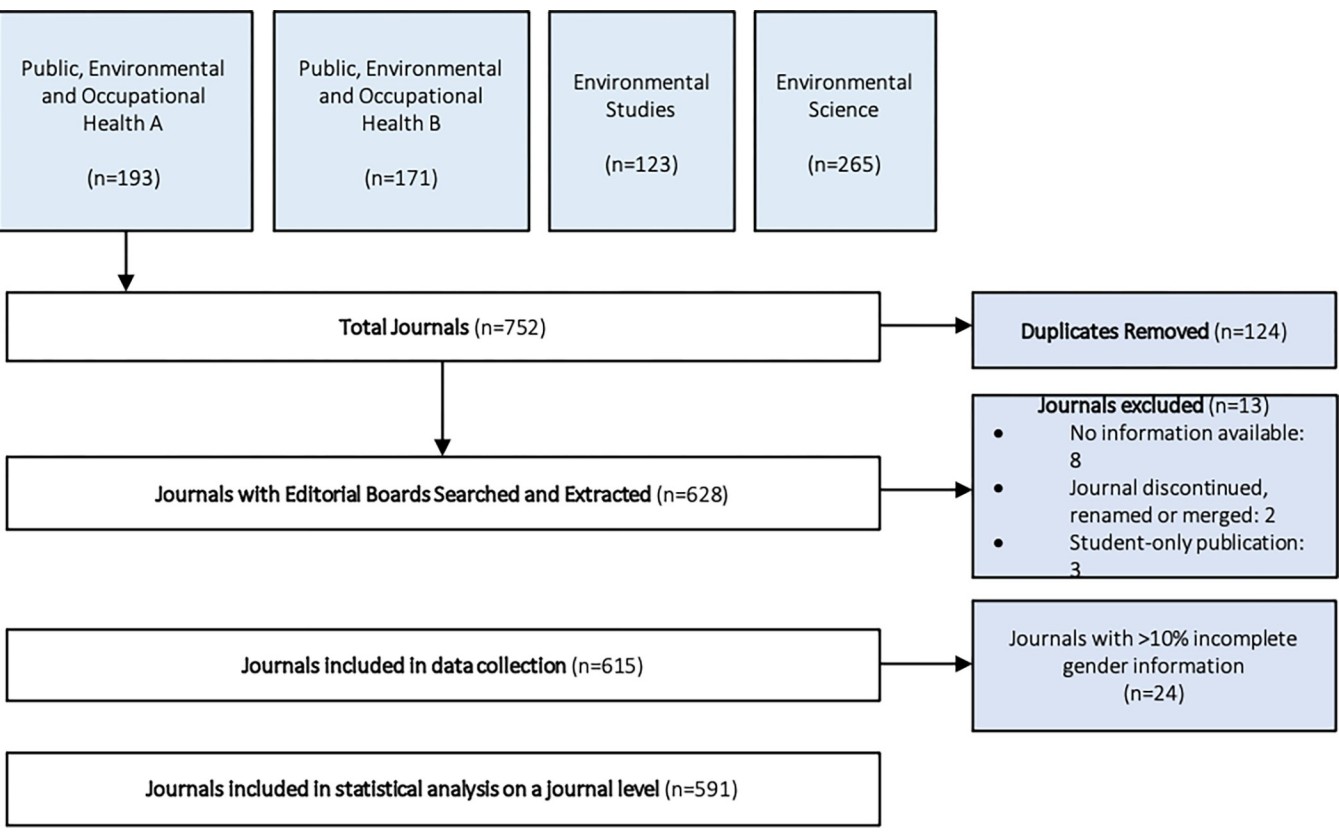

**Fig 1. Retrieval and selection of journals for statistical analysis.**

level of gender diversity with 51 (20.5%) journals displaying GP, 59 (23.7%) with a WGM majority and 139 (55.8%) with a majority of men (**Fig 2B**). Likewise, PEO health journals demonstrated a higher proportion of WGM EiC (35.6%) compared to journals classified as environmental (18.8%) or both (17.9%).

Geographically, the majority of editors' institutions were based in the UN Western Europe and Other region (76.9%), with 35.2% of editors (n = 9,761) coming solely from the United States, and 8.6% (n = 2,373) solely from the United Kingdom. This was followed by Asia and Pacific (14.1%) where 5.6% of the editors were from China, Latin America and the Caribbean (3.6%), Africa (2.7%), and Eastern Europe (2.6%). Overall, 9.8% of EiC, 15.8% of EL, 16.1% of EB, 15.3% of AB, 27.7% of EC, and 16% of all editors were LMIC-based. Only 25 journals (4.2%) had a majority of LMIC-based editors, while 15 (2.5%) demonstrated an equal presence of LMIC-based and HIC-based editors. Little difference between the different journal categories was observed in terms of economic diversity, with 85.7% of editors in PEO health journals, 82.7% of environmental journal editors, and 85.4% of editors in journals classified as both categories being HIC-based (**Fig 2A**). EiCs were predominantly based in the United States (35.5%), United Kingdom (14.0%), Canada (6.1%), Australia (5.5%), and China (4.0%). In total, 67.2% EiC were HIC-based men (n = 540), whilst only one EiC was a man based in a low-income country. In abject contrast, 27 (0.1%) editors in total were inferred WGM based in low-income countries. **S3–S5 Tables** provide in-depth details on the representation of each journal's editorial team by their editors' inferred gender, UN geographic region and socioeconomic status of the editors' institutions' countries following their JCR categories.

**Table 2. Summary of the data characteristics.**

| Data | |
|---|---|
| Journals | **N** |
| *Extracted from Clarivate* | 628 |
| *After exclusion* | 615 |
| *After removing journals with >10% incomplete data* | 591 |
| Editorial board roles (**total N = 27,722**) | **% (N)** |
| *Editor-in-Chief* | 2.9% (803) |
| *Editorial Leadership* | 16.9% (4,676) |
| *Editorial Board* | 86.8% (24,058) |
| *Advisory Board* | 13.2% (3,664) |
| *Early Career* | 0.4% (112) |
| Inferred gender editor (**total N = 27,722**) | **% (N)** |
| *Men* | 65.9% (18,280) |
| *Women* | 32.9% (9,127) |
| *Non-binary and other* | 0.05% (14) |
| *Missing* | 1.09% (301) |
| Inferred gender diversity of journal (**total N = 591**) | **% (N)** |
| *Majority WGM (>55% WGM)* | 11.3% (67) |
| *Gender parity (45–55% WGM)* | 13.2% (78) |
| *Majority men (>55% men)* | 75.5% (446) |
| Number of countries of editors' institutions | 150 |
| UN Region of editors' institution country (**total N = 27,722**) | **% (N)** |
| *Africa* | 2.7% (752) |
| *Asia and Pacific* | 14.1% (3,904) |
| *Eastern Europe* | 2.6% (721) |
| *Latin America and Caribbean* | 3.6% (1,010) |
| *Western Europe & Others* | 76.9% (21,324) |
| *Missing* | 0.04% (11) |
| Socio-economic status editors' institution country (**total N = 27,722**) | **% (N)** |
| *High-income* | 84% (23,280) |
| *Upper-middle income* | 11.8% (3,262) |
| *Lower-middle income* | 3.8% (1,040) |
| *Low income* | 0.4% (119) |
| *Missing* | 0.1% (21) |
| Economic diversity editors' institution country (**total N = 591**) | **% (N)** |
| *Majority High-income (>55% HIC)* | 93.2% (551) |
| *Economic parity (45–55% LMIC)* | 2.5% (15) |
| *Majority Low-and-middle income (>55% LMIC)* | 4.2% (25) |
| Journal Information (**total N = 591**) | |
| *Range of H-index* | 0–397 |
| *Range of impact factor* | 0.18–30.29 |
| *Range of Gender Inequality Index 2019* | 0.038–0.56 |
| *Range of editors per journal* | 5–736 |
| *Number of countries of journals* | 35 |
| *Number of publishers* | 164 |
| Range of Journal Gender Inequality Index 2019 (**total N = 591**) | |
| *By journal country* | 0.038–0.57 |
| *By editor-in-chiefs' country institution* | 0.025–0.57 |
| *By mean editorial leaderships' country institution* | 0.025–0.57 |
| *By mean editorial boards' country institution* | 0.049–0.57 |
| *By mean advisory boards' country institution* | 0.025–0.37 |
| *By mean overall editor country institution* | 0.056–0.57 |

**Fig 3** demonstrates variations in gender diversity across editors' institution countries (**A**), geographic region (**B**), and economic status (**C**). Further detail is available in **S6** and **S7 Tables**. Across the major journal publishers (publishers with >800 editors), low economic diversity

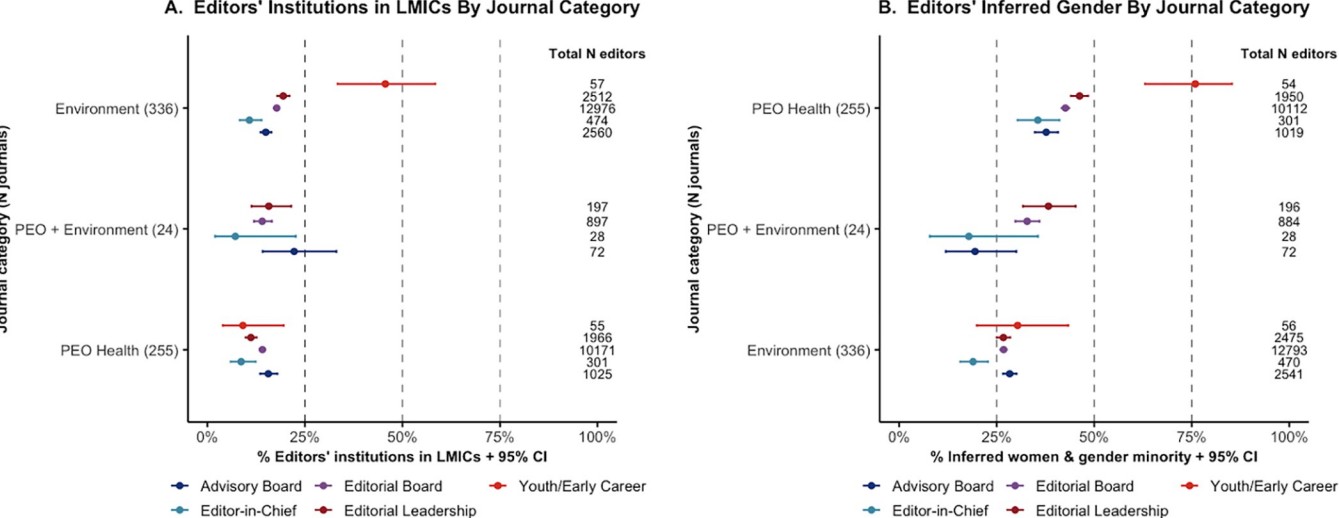

**Fig 2. Editors' inferred gender and editors' institution based in LMIC by journal JCR category.** Binomial 95% confidence intervals (CIs) were calculated for the % WGM and % editors LMIC-based. **A.** Editors' institution in LMICs by journal JCR category. **B.** Editors' inferred gender by journal JCR category. *Abbreviations*: LMICs: low- and middle-income countries; JCR: Journal Citation Reports; PEO: Public, Environment and Occupational; WGM: women and gender minorities.

was observed; none of the publishers showed parity in representation of HIC-based and LMIC-based editors (**Fig 4A**). Although various publishers displayed more gender diversity, only one publisher was considered to have gender parity (Routledge) (**Fig 4B**). Most journals were based in the United Kingdom (38.7%), United States (26.6%), Netherlands (12.9%), and Germany (4.7%). Journals based in LMICs tended to show a higher representation of LMIC-based editors compared to journals based in HICs (**Fig 5A**). No clear trend with the GII of the journal country and gender diversity across their editors could be observed (**Fig 5B** and **S8 Table**).

**Fig 6A** and **6B** visualizes where journals fall in terms of both the gender diversity and economic diversity of their editors. Whilst some journals show parity in either gender or economic diversity, most journals are located in the bottom left quadrant and have a majority of men and a majority of HIC-based editors. Only one journal (out of 591) demonstrated both gender and economic parity (AJAR-African Journal of AIDS Research). The limited number of journals categorized in both JCR categories (**Fig 6A, red**) all consisted of HIC-based editor majorities. When exploring the same visualization at the level of countries journals are published in (**Fig 6C**), journals from Western Europe and Other countries showed little inclusion of LMIC-based editors. The Malawi Medical Journal, published in Malawi, included 100% of LMIC-based editors, as most of their editors were affiliated with institutions in Malawi. Five countries of published journals showed overall gender parity (Ireland, Australia, Spain, Brazil, Argentina).

## Correlation between variables

Correlations between the journal-based indicators analysed are shown in **Fig 7**. A small negative correlation between the percentage of LMIC-based and percentage of WGM is observed for both editorial teams overall (R = -0.17) and for editor sub-roles (orange). Having WGM and LMIC-based EiCs are both positively correlated to overall percentage of WGM (R = 0.42) and overall percentage of LMIC-based (R = 0.57) editors respectively (purple). Likewise, mean GII of EiCs is positively correlated with mean GII of all editorial board members (R = 0.7,

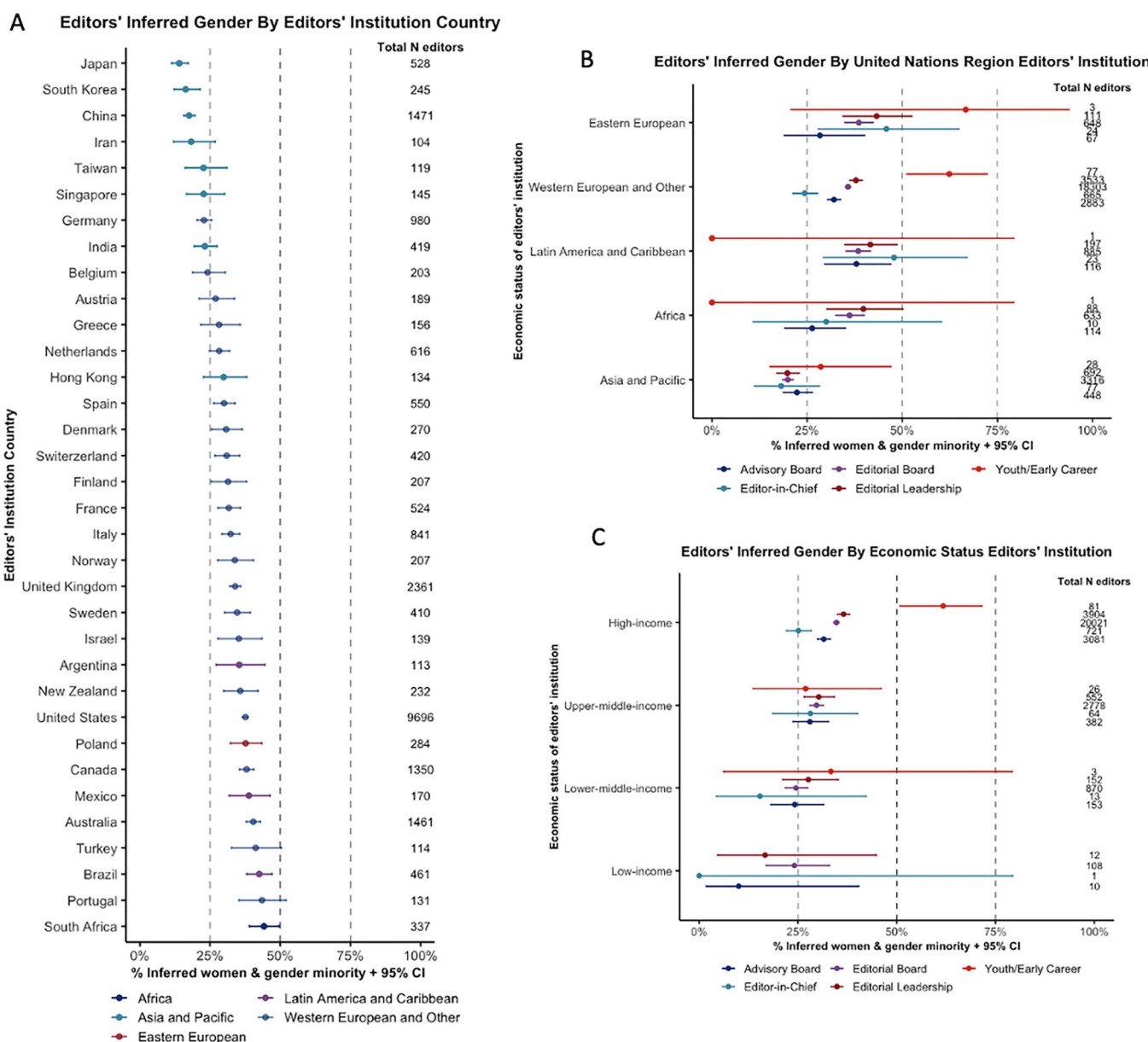

**Fig 3. Editors' inferred gender by editors' institution categories.** Binomial 95% confidence intervals (CIs) were calculated for % Women & Gender Minorities. **A.** Editors' inferred gender by editors' institution country. Only countries with >100 editors are presented (n = 34). **B.** Editors' inferred gender by the United Nations region category of the editors' institution and editor sub-roles. **C.** Editors' inferred gender by economic status editors' institution and editor sub-roles.

purple). Furthermore, the GII of the journal country is positively correlated with the mean GII of the editors' institution countries (R = 0.61, yellow). Positive correlations can also be observed between the GII of the journal country and mean GII across editor sub-roles with the percentage of LMIC-based across several editor sub-roles (blue). In contrast, non-significant or very small correlations are observed between the GII of the journal country and mean GII across editor sub-roles with the percentage of WGM across editor sub-roles (green). The high intercorrelation between editor roles metrics follows naturally from the fact that a proportion of editors are included in multiple role groups.

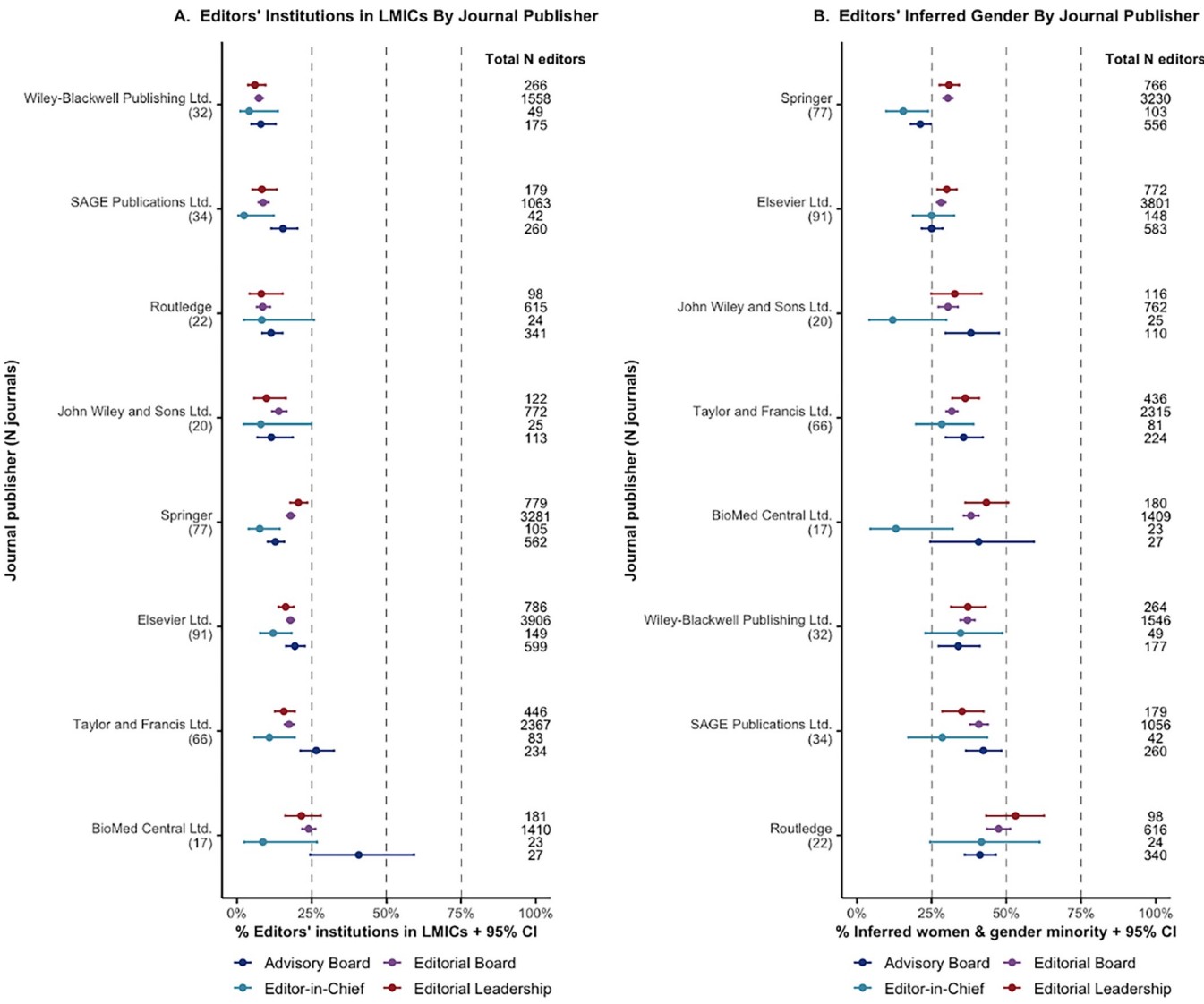

**Fig 4. Editors' inferred gender and editors' institution based in LMIC by journal publisher and editor sub-roles.** Binomial 95% confidence intervals (CIs) were calculated for the % WGM and % editors LMIC-based. The journal publishers with most editors affiliated (>800 editors) in the dataset were included (n = 8). The youth/early career researcher role was omitted from this figure. **A.** Editors' institution in LMICs by journal publisher and editor sub-roles. **B.** Editors' inferred gender by journal publisher and editor sub-roles. *Abbreviations:* LMICs: low- and middle-income countries; WGM: women and gender minorities.

## Regression analysis

**S1 Fig** provides further visualization using simple linear regression of the observed trends across the different JCR journal categories at a journal level. Corresponding to observed correlations, whilst positive trends could be observed between overall percentage of WGM and percentage of WGM in leadership (**A**), and between mean GII of the EB and GII of the journal country (**D**), a negative trend could be observed between the percentage of overall WGM and percentage of overall LMIC-based editors (**C**). GII of the journal country and overall percentage of WGM displayed slight positive trends across the journal categories.

When plotting editors' journal country against the GII of the journal country and the percentage of WGM, a slight negative trend in a simple linear regression (unweighted) can be

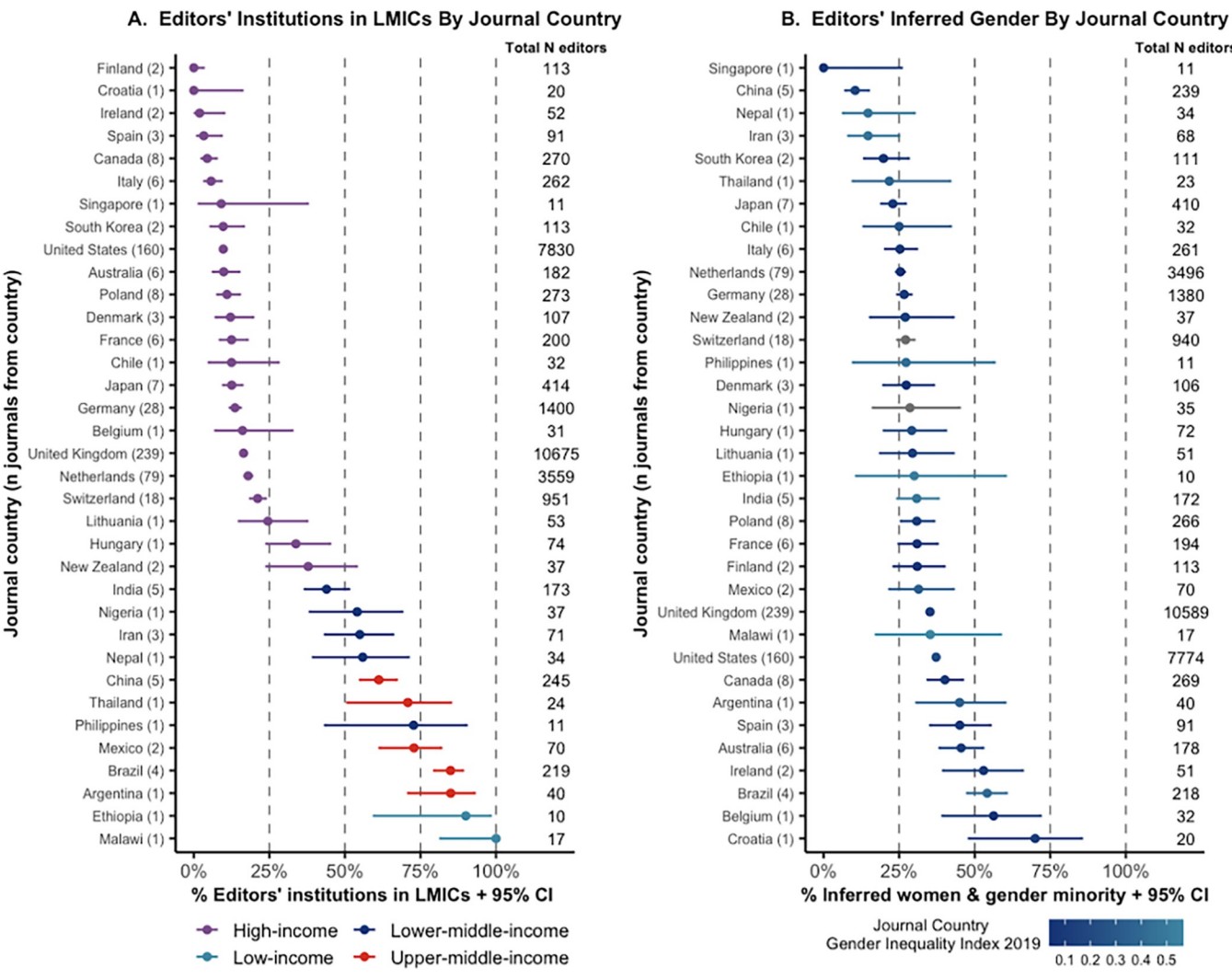

**Fig 5. Editors' inferred gender and editors' institution based in LMIC by country that the journal is published in (journal country).** Binomial 95% confidence intervals (CIs) were calculated for the % WGM and % editors LMIC-based. **A.** Editors' institution in LMICs by journal country. **B.** Editors' inferred gender by journal country displayed with the gender inequality index (2019) for the journal country. *Abbreviations:* LMICs: low- and middle-income countries; WGM: women and gender minorities.

observed (**Fig 8A**). Similarly, **Fig 8B** demonstrates a negative trend in simple linear regression (unweighted) on a journal country level. Lastly, when exploring the relation between the GII of the EiC with the mean GII of the overall editorial team, a positive trend can be observed (**Fig 9**).

Further, linear regression (on journal level) and logistic regression (on editor level) analyses were run to explore the influence of different journal variables on the presence of WGM and LMIC-based editors in journal editorial teams (**Table 3**). As shown in **Model A**, as the journal %WGM EiCs increase, the percentage of WGM in editorial teams increase. Yet, in contrast to our hypothesis, when the country GII (gender inequality) increases, the percentage of WGM in editorial teams increases. In further sensitivity analyses, this seems to be explained by the high number of editors based in the United States (n = 9,761); a country with a relatively high GII whilst having relatively high gender diversity among its editors compared to other countries in the same GII range (**see Model B**). This effect could similarly be observed when

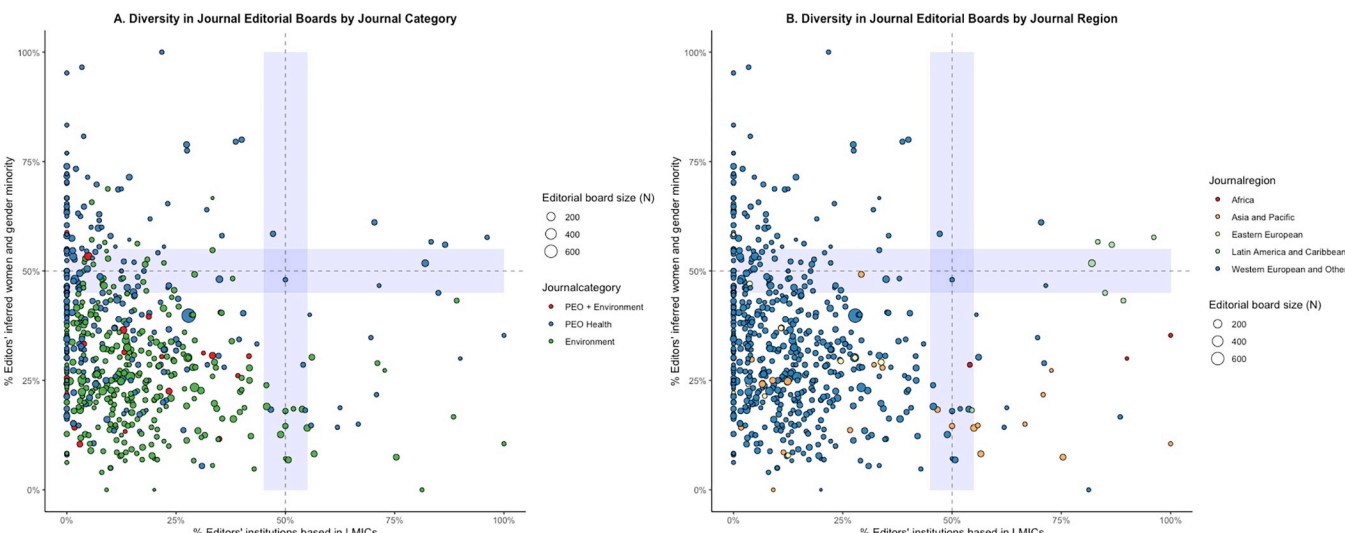

**Fig 6. Editors' inferred gender and editors' institution based in LMIC by journal and journal country.** Blue-shaded areas indicate gender parity (45–55% women) and economic parity (45–55% LMIC-based). **A.** Editors' diversity at a journal level colored by journal category. **B.** Editors' diversity at a journal level colored by region of the journal. **C.** Editors' diversity by journal country colored by journal UN region. ***Abbreviations:*** LMICs: low- and middle-income countries; UN: United Nations.

running logistic regression analyses on an editor level (**Model, E, F**). As shown in **Model C,** when the GII of the journal country increases, the percentage of LMIC-based editors likewise increases. This effect is slightly higher when excluding editors based in the United States (**Model D**) and is similar in observation to the logistic regression models on LMIC-based editors (**Model G, H**).

## Discussion

This analysis exemplifies a lack of inclusion in LMIC-based researchers and women and gender minorities in editorial teams of journals in the field of environmental sciences and PEO health. Over 65% of editors included in this analysis were inferred as men while more than three-quarters of editors were affiliated with institutions in the Western Europe and Other UN region. Reflecting this, the majority of journals (70.2%, n = 415) had both majority men and majority HIC-based editors, with only one journal demonstrating gender and economic parity. Additionally, 67.2% of EiCs were HIC-based men (n = 540), whilst no EiC was a low-income based WGM. Journals with EiCs that were inferred as WGM or from institutions based in LMICs tended to have more gender and economic diversity. These findings reflect similar reports across literature on the stark representation gaps in academia and publishing [10–14,16–24,37]. For example, the UNESCO 2021 Science Report exhibits the global share of scientific publications across income groups and regions in 2019 –reporting 62.9% of publications from HICs, with East & Southeast Asia (36.7%), Europe (34.9%), and North America (23.2%) as the most represented regions [37].

In addition to influencing the publications accepted and progression/evolution of journals, editorial teams reflect existing power structures/imbalances in academia that often mirror the influences of colonialism [18,23,28,38]. As a result of this colonial legacy, the direction of knowledge flow often appears one-sided and therefore misses out on the varied perspective and experiences of vast majorities of the global population [38,39]. In the case of environmental sciences, public health, and planetary health topics, more diverse, just and equitable systems to produce and disseminate knowledge would also enable more comprehensive and

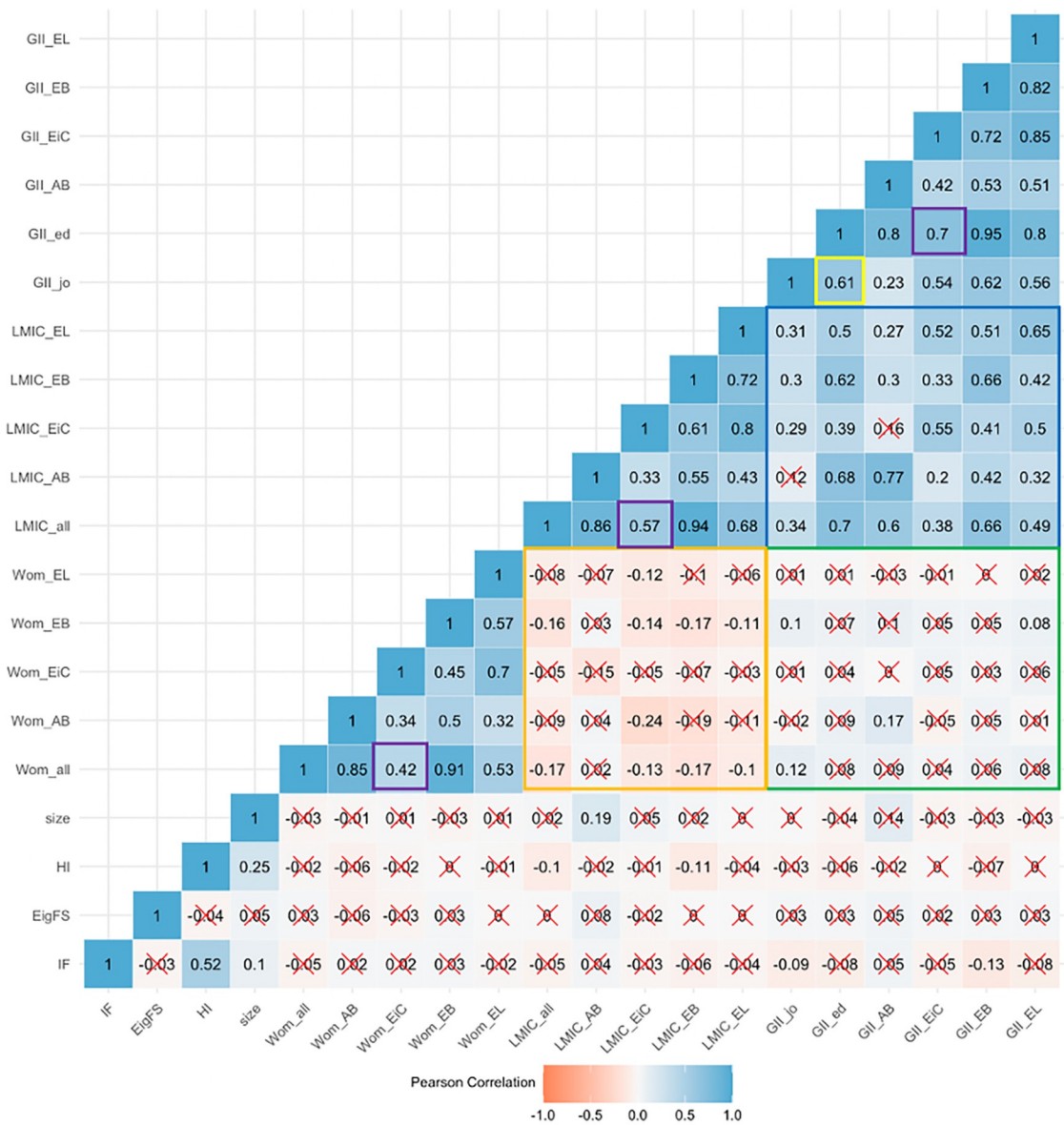

**Fig 7. Correlations between journal-based indicators, women and gender minority presence, LMIC presence and GII indexes by editor role. X;** not statistically significant on alpha = 0.05 level; An FDR adjustment is used to address multiple testing. **GGI_EL**; average gender inequality index of editorial leaderships' institutions, **GGI_EB**; average gender inequality index of editorial boards' institutions, **GGI_EiC**; average gender inequality index of editor in chiefs' institutions, **GGI_AB**; average gender inequality index of advisory boards' institutions, **GII**: gender inequality index of journal country, **LMIC_EL**: percentage of editorial leaderships' institutions based in LMIC, **LMIC_EL**: percentage of editorial leaderships' institutions based in LMIC, **LMIC_EB**: percentage of editorial boards' institutions based in LMIC, **LMIC_EiC**: percentage of editor-in-chiefs' institutions based in LMIC, **LMIC_AB**: percentage of advisory boards' institutions based in LMIC, **LMIC_all**: percentage of all editors institutions based in LMIC, **Wom_EL**: percentage of editorial leadership inferred women or gender minority, **Wom_EL**: percentage of editorial leadership inferred women or gender minority, **Wom_EB**: percentage of editorial board inferred women or gender minority, **Wom_EiC**: percentage of editor-in-chief inferred women or gender minority, **Wom_AB**: percentage of advisory board inferred women or gender minority, **Wom_all**: percentage of all editors inferred women or gender minority, **Size**; size of the editorial board, **HI**; h-index of journal, **EigFS**; eigen factor score of journal, **IF**; impact factor of journal.

multidisciplinary approaches to addressing these challenges that inherently have varying degrees of global and justice implications [38]. For example, climate change reveals deeper questions around justice as it interacts with existing social and economic inequalities,

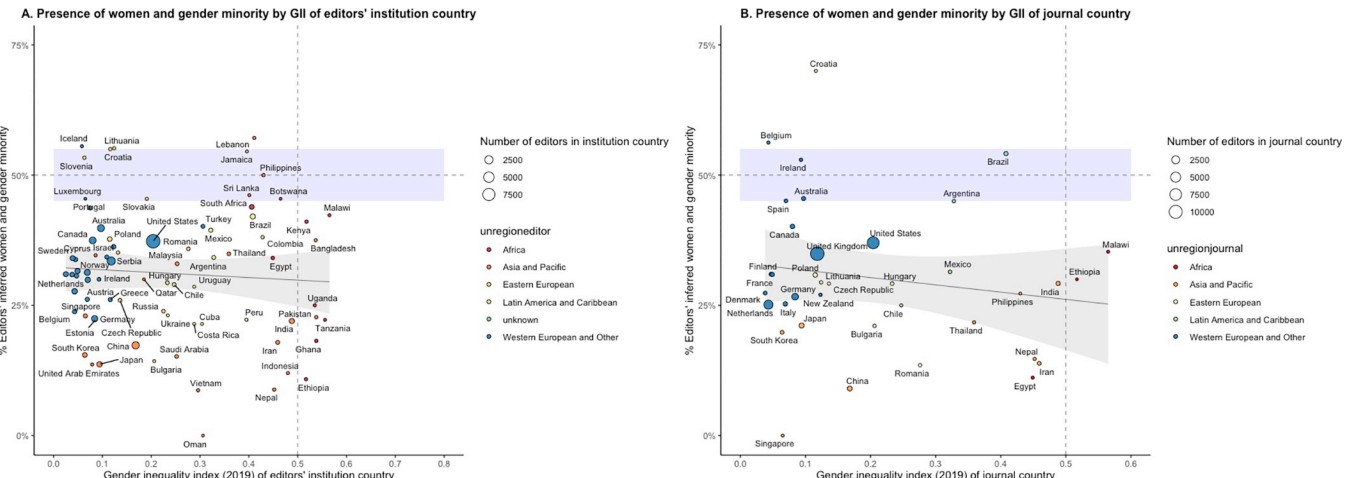

**Fig 8. Editors' inferred women and gender minority by gender inequality index (2019) of journal country.** Blue-shaded areas indicate gender parity (45–55% women). Simple linear regression (unweighted) with 95% confidence interval (CI) is depicted between % editors' inferred WGM and GII of journal country. **A.** Editors' institution country is plotted against % editors inferred WGM and GII of journal country. Editors' institution countries with less than 10 editors per country were excluded. **B.** Journal country is plotted against % editors inferred WGM and GII of journal country. *Abbreviations:* LMICs: low- and middle-income countries; WGM: women and gender minorities; GII: gender inequality index.

exacerbating long-standing trends within and between countries where those that contribute the least are often impacted the most.

Similar studies have demonstrated lower proportions of women in first authorship positions in high-impact journals, while less than a quarter of commissioned commentaries in prestigious journals are authored by women [40–42]. The UNESCO report highlights that in most countries, the majority of tertiary degree graduates in health and the natural sciences are women [37]. Still, women are significantly underrepresented in first-author publications, editorial teams, leadership positions, and specifically EiC roles. Additionally, the majority of community health workers/representatives are women, including women of colour [43,44]. Yet while women conduct a significant portion of global health work on the ground, they are not equally represented in global health leadership positions nor in the editorial boards of scientific journals [45,46].

These patterns may be explained in part by the social phenomenon, homophily, which indicates that people are more likely to work with and share networked information with peers who share similar characteristics to them. In the context of existing global power dynamics, this phenomenon could promote segregation and stunt scientific advancement [47,48]. Additionally, editorial service is also considered in tenure reviews, contributing to career advancement in academia. This means an imbalance of representation on editorial teams has an impact beyond what gets published, but also women's career progressions. A number of factors may play a role in influencing this dearth of women on editorial teams, including the challenges faced by women academics in career progression such as lack of women mentors, and institutional sexism [42,49,50]. For example, this became particularly apparent in the first months of the COVID-19 pandemic as women often faced an imbalanced burden of domestic labor [42]. Furthermore, women more often take on higher loads of unpaid work that include not only family responsibilities and care duties, but also academic roles or 'career development' tasks [51–53]. In terms of geographic representation, it is worth noting that the vast majority of science is shared in English which may serve as an obstacle to the involvement of academics who are not native-English speakers [49,54,55].

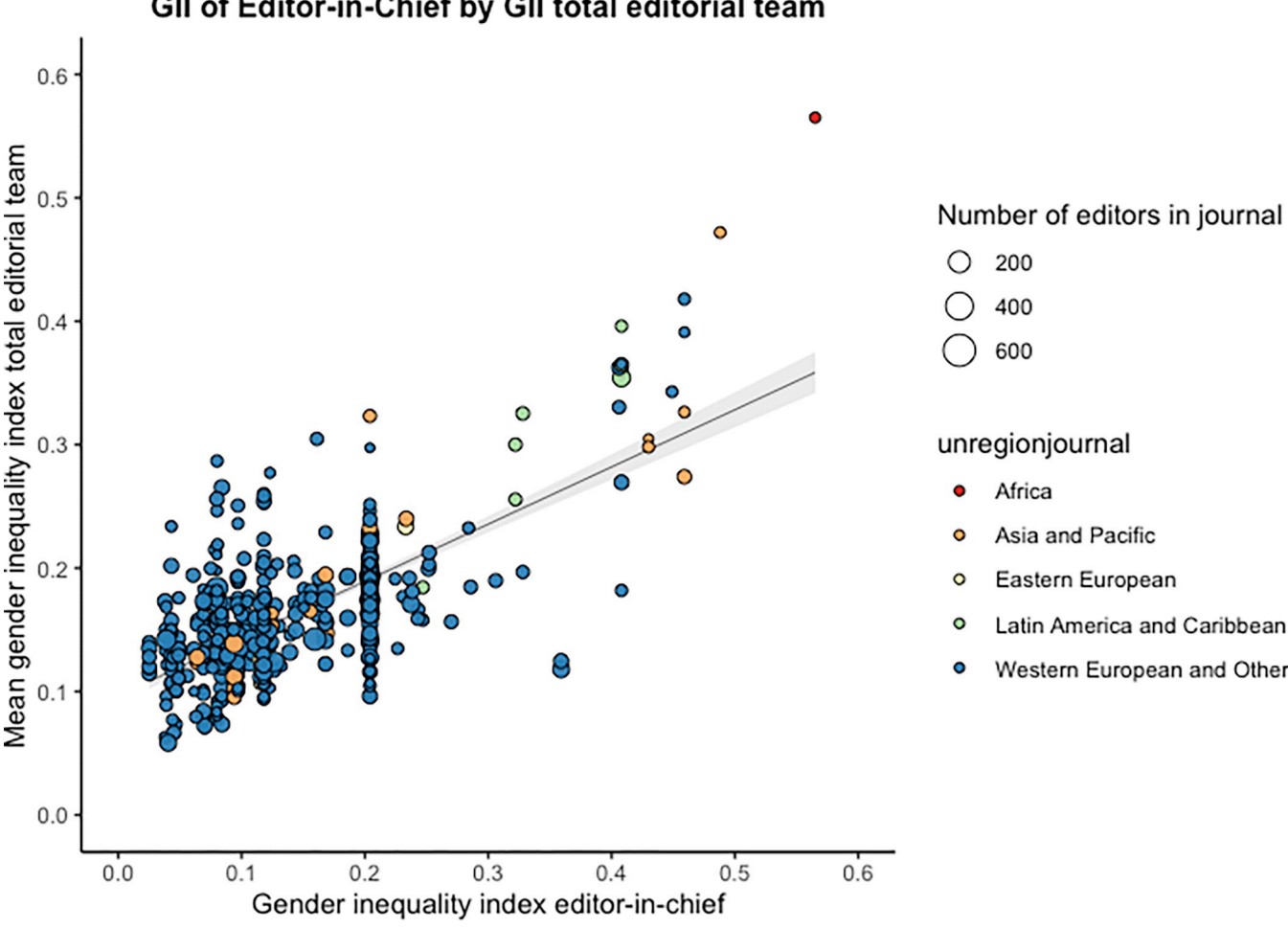

**Fig 9. Editorial teams' mean Gender Inequality Index (GII) by GII of editor-in-chief.** Simple linear regression (unweighted by the number of editors in the journal) with 95% confidence interval (CI) is depicted.

A number of policies and strategies have been suggested to ensure or promote both gender and geographic diversity in academia including transparency and opportunities for exposure to relevant professional and academic experience. For example, journals can improve transparency by publicizing the representation of their editorial teams. This has already been implemented by a number of journals included in this study, such as the publisher Elsevier, which recently released a report, "The Researcher Journey Through a Gender Lens," as part of their efforts to advance gender diversity and equity in research [19,56,57]. This transparency provides a mechanism to hold journals accountable for their editorial teams decisions. This may also apply to the leadership of EiCs in their recruitment strategies for editorial team members. EiCs, or leaders, that actively recruit WGM and academics from institutions outside of HICs are essential to ensuring both gender and geographic parity rather than perpetuating the cycle of 'the old boys club' that is often present. Journal management could consider gender and geographic quotas to maintain a minimum percentage of these groups on their editorial teams. Another avenue for improving accountability can come from funding. First, there is a gulf between how academic labour (such as peer review) is valued and the profit it brings to publishers. Additional transparency as well as remuneration for academic labour is worthwhile for encouraging diversity [58]. Second, by requiring equitable collaborations, funders can

**Table 3. Summary of regression analyses.**

*Journal level analysis*

| A | Linear regression | | B | Linear regression | | C | Linear regression | | D | Linear regression | |
|---|---|---|---|---|---|---|---|---|---|---|---|
| *Independent variables* | β Coefficient | Std. error | | β Coefficient | Std. error | *Independent variables* | β Coefficient | Std. error | | β Coefficient | Std. error |
| Editorial board size | -0.0005* | 0.0002 | | -0.0005 | 0.0003 | Editorial board size | 0.000159 | 0.000251 | | -0.000242 | 0.000373 |
| % WGM editor-in-chief | 0.173*** | 0.016 | | 0.186*** | 0.019 | % LMIC editor-in-chief | -0.027 | 0.017 | | 0.303*** | 0.035 |
| Journal country GII | 0.256** | 0.086 | | 0.105 | 0.101 | Journal country GII | 0.703*** | 0.090 | | 0.769*** | 0.119 |
| Journal impact factor | -0.0025 | 0.0026 | | -0.0022 | 0.0029 | Journal impact factor | -0.00028 | 0.00273 | | 0.002244 | 0.002584 |

*Editor level analysis*

| E | Logistic regression | | F | Logistic regression | | G | Logistic regression | | H | Logistic regression | |
|---|---|---|---|---|---|---|---|---|---|---|---|
| *Independent variables* | OR | 95% CI | | OR | 95% CI | *Independent variables* | OR | 95% CI | | OR | 95% CI |
| Editorial board size | 0.999** | 0.998–1.000 | | 0.998*** | 0.997–0.999 | Editorial board size | 0.999 | 0.997–1.001 | | 1.006*** | 1.002–1.009 |
| % WGM editor-in-chief | 1.983*** | 1.862–2.112 | | 1.988*** | 1.830–2.159 | % LMIC editor-in-chief | 7.459*** | 6.051–9.184 | | 3.471*** | 2.557–4.691 |
| Journal country GII | 1.420** | 1.103–1.836 | | 0.954 | 0.727–1.249 | Journal country GII | 3.100e+10*** | 1.029e+10–9.856e+10 | | 8.189e+19 | 6.208e+18–1.209e+21 |
| Journal impact factor | 0.979*** | 0.968–0.990 | | 0.969*** | 0.955–0.983 | Journal impact factor | 1.084*** | 1.062–1.105 | | 1.088*** | 1.060–1.116 |

Dependent variable: % WGM of editors  Dependent variable: % LMIC-based editors.

**A:** model including editors from the United States  **C:** model including editors from the United States.

**B:** model excluding editors from the United States  **D:** model excluding editors from the United States.

* *P*-value < 0.05, ** *P*-value < 0.01, *** *P*-value < 0.001. * *P*-value < 0.05, ** *P*-value < 0.01, *** *P*-value < 0.001.

Dependent variable: being a woman or gender minority  Dependent variable: being a LMIC-based editor.

**E:** model including editors from the United States  **G:** model including editors from the United States.

**F:** model excluding editors from the United States  **H:** model excluding editors from the United States.

* *P*-value < 0.05, ** *P*-value < 0.01, *** *P*-value < 0.001 * *P*-value < 0.05, ** *P*-value < 0.01, *** *P*-value < 0.001.

**Abbreviations:** WGM: Women and gender minorities; LMICs: Low- and middle-income countries; GII: Gender inequality index.

influence how projects are structured and therefore the gender and regional representation of academic outputs. The geosciences and environmental science fields were amongst the leading sectors for publications from international collaborations in 2019 [37]. In the long run, these approaches could open the door for partnerships and funding support for researchers and hence promote gender equality in academia [59,60].

Exposure to relevant professional experience through dedicated resources, administrative support, mentorship, and clear advancement pipelines to support women, gender minorities, and individuals from low- and lower-middle income countries can also provide an avenue to improve editorial board diversity. In terms of the monopoly of English on scientific publishing, journals and institutions must prioritize multilinguistic opportunities (such as editorial support or increasing the diversity of languages in publishing) for researchers that would encourage not only the publication of literature from non-native English speakers but also their involvement in journals' editorial boards [55]. Early access to mentorship and career development may improve the retention and success of women and gender minorities in academia, including connecting them to future opportunities such as editorial roles [61]. As a

result, it may be worth considering the representation of different generations on knowledge creation and dissemination.

One programme implemented by 13 of the included journals involves 'early career editorial boards' composed of young researchers/professionals. These EC boards, implemented differently across journals, contribute to the journal through the identification of research and topics while also serving as a mechanism for inclusion and a professional pipeline that provides mentorship opportunities for young professionals to eventually take on editorial leadership roles [62,63]. In this way, EC boards benefit not only young researchers' career progression but the advancement of the journal as well. In some cases, these EC boards had higher proportions of women and gender minorities. It will be interesting to consider if this "pipeline" that seems closer to achieving gender parity is then reflected in the future editorial teams of these journals. However, this did not correspond to the geographic or socioeconomic distribution/ representation. This is exemplified by the fact that while there were 81 (of 110) early career editorial board members from HICs and 26 from upper-middle-income countries, there were only three individuals from LMICs and none from low-income countries.

## Strengths and limitations

Two of the major strengths of this study are the broad transdisciplinary scope and large sample size of 591 journals with 27,722 editors. By analysing all journals in the selected JCR categories, this study provides the first comprehensive view of the current state of academic diversity amongst editorial teams in environmental sciences and PEO health on a global scale. Furthermore, an important strength of this study is that the research team was largely composed of a diverse group of early career researchers which fostered opportunities for equal collaboration and skills development [64,65]. However, this study also has some limitations. Firstly, academia is also characterized by power imbalances across race and ethnicity, but it was not possible to infer race and ethnicity from biographies. While this analysis provides insight into the representation of gender and geography, this is only the first step. It is important for future work (both research and journal editorial board policies) to also consider ethnic, linguistic, socioeconomic, and racial minorities and to ensure that these perspectives are not further marginalized in the processes of knowledge creation and dissemination. Secondly, in this study we primarily used publicly available information to infer gender. This approach was chosen to prevent the potential bias induced by gender-to-name algorithms that are unable to identify people outside the gender binary and are less efficient in inferring gender from non-Western names. However, in spite of these limitations, a gender-to-name algorithm was used for 7.6% (n = 2,170) of individuals where it was not possible to find publicly available gender information. It is important to acknowledge the inherent stigma and, in some cases, danger attached to inferring as a gender minority in different parts of the world influencing people's ability, freedom, and safety to openly identify as who they are. Future studies may wish to build on our findings by using surveys that allow people to self-identify their gender, race, and ethnicity.

## Conclusion

This study demonstrates an evident imbalance in gender and geographic representation of journal editorial teams publishing in the intersection of environmental sciences and public health. The cross-sections of these fields, such as the field of planetary health, call for interdisciplinary, comprehensive and global solutions. Yet, how could one envision such solutions when a select, unrepresentative demographic controls the voice and narrative of what is considered scientific knowledge? By drawing attention to the current disparity in representation, this

study exposes the work needed to be done and suggests key recommendations to make progress moving forward.

## Supporting information

**S1 Fig. Simple linear regression analyses by journal category.**
(DOCX)

**S1 Table. Excluded journals with reason for exclusion.**
(DOCX)

**S2 Table. Journals excluded due to incomplete inferred gender data.**
(DOCX)

**S3 Table. Characteristics of journals categorized as Public, Environmental and Occupational Health Journals following the Journal Citation Reports (JCR).**
(DOCX)

**S4 Table. Characteristics of journals categorised as Environmental Sciences and Environmental Studies following the JCR.**
(DOCX)

**S5 Table. Characteristics of journals categorised as Public, Environmental and Occupational Health AND Environmental Sciences and Environmental Studies following the JCR.**
(DOCX)

**S6 Table. Inferred gender composition of editorial boards by World Bank income group and United Nations (UN) geographic region of editors' institutions.**
(DOCX)

**S7 Table. Inferred women and gender minority composition of editorial boards categorised by UN region and World Bank income group of editors' institutions.**
(DOCX)

**S8 Table. Representation of editors by country of journals.**
(DOCX)

## Acknowledgments

The authors would like to gratefully thank Richard Arthur John Dear for providing insightful discussions and expertise for the data-analysis in R. The authors thank the peer-reviewers for their constructive feedback that enabled the improvement of this manuscript.

## Author Contributions

**Conceptualization:** Sara Dada, Kim Robin van Daalen.

**Data curation:** Sara Dada, Kim Robin van Daalen, Alanna Barrios-Ruiz, Kai-Ti Wu, Aidan Desjardins, Mayte Bryce-Alberti, Alejandra Castro-Varela, Parnian Khorsand, Ander Santamarta Zamorano, Laura Jung, Grace Malolos, Jiaqi Li, Dominique Vervoort, Nikita Charles Hamilton, Poorvaprabha Patil, Omnia El Omrani, Telma Sibanda.

**Formal analysis:** Kim Robin van Daalen.

**Funding acquisition:** Kim Robin van Daalen.

**Methodology:** Sara Dada, Kim Robin van Daalen.

**Project administration:** Sara Dada, Kim Robin van Daalen.

**Visualization:** Kim Robin van Daalen.

**Writing – original draft:** Sara Dada, Kim Robin van Daalen.

**Writing – review & editing:** Sara Dada, Kim Robin van Daalen, Alanna Barrios-Ruiz, Kai-Ti Wu, Aidan Desjardins, Mayte Bryce-Alberti, Alejandra Castro-Varela, Parnian Khorsand, Ander Santamarta Zamorano, Laura Jung, Grace Malolos, Jiaqi Li, Nikita Charles Hamilton, Poorvaprabha Patil, Omnia El Omrani, Marie-Claire Wangari, Telma Sibanda, Conor Buggy, Ebele R. I. Mogo.

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
