## [Decision Letter · Decision Letter 0]

12 Apr 2022

PGPH-D-21-01099

Challenging the “Old Boys Club” in Academia: Gender and Geographic Representation in Editorial Boards of Journals Publishing in Environmental Sciences and Public Health

Dear Ms. Dada, 

Thank you for submitting your manuscript to PLOS Global Public Health. After careful consideration, we feel that it has merit but does not fully meet PLOS Global Public Health’s publication criteria as it currently stands. Therefore, we invite you to submit a revised version of the manuscript that addresses the points raised during the review process.

Please address comments from both reviewers on your revision round. Some details need further elaboration as suggested by the reviewers. 

The typographical suggestions also need to be taken into account. 

Additionallty, the percentages in Table 2, Editorial board roles section do not add up. Please check this out as well. 

We look forward to receiving your revised manuscript.

Kind regards,

Zahra Zeinali, MD MPH

Academic Editor

Journal Requirements:

1. Your co-authors:

Laura Jung -laura.kunignde@gmail.com

Jiaqi Li -jl2006@cam.ac.uk

Ebele R. I. Mogo -ebele@erimconsulting.org

,have not confirmed authorship of the manuscript. We have resent them the authorship confirmation email; however please check that the above email address for them is correct and follow up personally to ensure they confirm. 

Please note that we cannot proceed your manuscript  until we have received confirmations from all co-authors.

2. Please amend your detailed Financial Disclosure statement. This is published with the article, therefore should be completed in full sentences and contain the exact wording you wish to be published.

i) Please include all sources of funding (financial or material support) for your study. List the grants (with grant number) or organizations (with url) that supported your study, including funding received from your institution. 

ii). State the initials, alongside each funding source, of each author to receive each grant.

iii). State what role the funders took in the study. If the funders had no role in your study, please state: “The funders had no role in study design, data collection and analysis, decision to publish, or preparation of the manuscript.”

3. Please provide  separate figure files in .tif or .eps format only and remove any figures embedded in your manuscript file.  Please ensure that all files are under our size limit of 20MB.  

For more information about how to convert your figure files please see our guidelines: Once you've converted your files to .tif or .eps, please also make sure that your figures meet our format requirements

4. In the online submission form, you indicated that [Insert text from online submission form here]. All PLOS journals now require all data underlying the findings described in their manuscript to be freely available to other researchers, either 1. In a public repository, 2. Within the manuscript itself, or 3. Uploaded as supplementary information.

5. Please update the completed 'Competing Interests' statement, including any COIs declared by your co-authors. If you have no competing interests to declare, please state "The authors have declared that no competing interests exist"

6. We have noticed that you have uploaded supporting information but you have not included a list of legends.  Please add a full list of legends for all supporting information files (including figures, table and data files) after the references list. 

Additional Editor Comments (if provided):

Dear authors,

Thank you for submitting this interesting study to PLoS GPH.

We deem this submission relevant to the scope of our journal and would be happy to publish it after some minor revisions.

We look forward to the next revision of this study,

Kind regards,

Dr. Zeinali

Reviewers' comments:

Reviewer's Responses to Questions

**Comments to the Author**

1. Does this manuscript meet PLOS Global Public Health’s publication criteria? Is the manuscript technically sound, and do the data support the conclusions? The manuscript must describe methodologically and ethically rigorous research with conclusions that are appropriately drawn based on the data presented.

Reviewer #1: Yes

Reviewer #2: Yes

2. Has the statistical analysis been performed appropriately and rigorously?

Reviewer #1: Yes

Reviewer #2: I don't know

3. Have the authors made all data underlying the findings in their manuscript fully available (please refer to the Data Availability Statement at the start of the manuscript PDF file)?

Reviewer #1: Yes

Reviewer #2: Yes

4. Is the manuscript presented in an intelligible fashion and written in standard English?

Reviewer #1: Yes

Reviewer #2: Yes

5. Review Comments to the Author

Reviewer #1: I congratulate the authors on a well-researched and well-written article. The topic is timely given ongoing interest in the power dynamics surrounding the construction of knowledge, especially in the field of global public health. I offer two key areas for thought related to the methods and discussion.

Methods: (1)How did the author's address individuals who had multiple affiliations--perhaps an affiliation at universities both in LMICs and HICs (as can be common among GH practitioners)? (Lines 200-203).

(2) Why were student-run editorial boards excluded (lines 181-182)? Quick reflection on such data could reveal prospects for changes for the future or offer a data point for similar studies to be conducted in the future.

Discussion: (1) It is important to discuss the role that women play in global public health, especially on-the ground. Community Health Workers/Representatives are overwhelmingly female (see Love 1997, and more recent numbers from WB). This means that we depend on women (and many women of color) to carryout policies and programming, but they are not equally represented in editorial boards. Inclusion of such information in your discussion only strengthens your argument on the production of knowledge.

(2)More discussion of the limitation of not considering ethnicity would also strengthen the piece. Though I completely agree with the decision not to include in the present analysis, some discussion of how editorial staff from LMICs is only a first step. Those from LMICs likely do not represent ethnic/linguistic/racial minorities. This is another part of the ongoing legacy of the coloniality of global public health.

I also offer minor typographical errors or stylistics choices for the authors to consider:

Lines 138-139: "field's ability to...improved health for all" This conclusion seems to be a jump from the evidence presented in this paragraph. Add 1-2 sentences of evidence or considering moving to the next paragraph where it seems to more accurately reflect the information presented.

Line 152: Consider adding male to the White, Western lens

Line 163: the phrase "contributes to shaping tomorrow" is accurate but feels trite, consider revision

Line 180: remove "would"

Lines 194-196: Should these journal titles be italicized in text?

Lines 208-210: Remove GII index explanation- no other metrics are explained in this section

Line 255: Capitalize I in Confidence Interval abbreviation

Line 494: Insert % (and number of individuals) for whom the gender-to-name algorithm was used

Throughout results & discussion: I am unsure of the style guide's preference for using % to mean percent when no number is offered. For example, % overall WGM (line 360) versus percent overall WGM.

Reviewer #2: This is a compelling and clearly written analysis of editorial boards for journals in the environmental sciences and public health space. The multiple authors have detailed their methods and impressive efforts in compiling a unique database of editorial board members for the relevant journals.

Writing is excellent overall.

Tables and figures are compelling and emphasize important results.

Two minor typographical comments:

Line 109 - 'published in as scientific' - should be "published as scientific"

Line 494 - "for X% of individuals" - should be replaced with the final percentage (7.6%?)

6. PLOS authors have the option to publish the peer review history of their article (what does this mean?). If published, this will include your full peer review and any attached files.

**Do you want your identity to be public for this peer review?** For information about this choice, including consent withdrawal, please see our Privacy Policy.

Reviewer #1: **Yes: **Meghan Farley Webb

Reviewer #2: No

---

## [Editor Report · Decision Letter 1]

9 May 2022

Challenging the “Old Boys Club” in Academia: Gender and Geographic Representation in Editorial Boards of Journals Publishing in Environmental Sciences and Public Health

PGPH-D-21-01099R1

Dear Ms. van Daalen,

We are pleased to inform you that your manuscript 'Challenging the “Old Boys Club” in Academia: Gender and Geographic Representation in Editorial Boards of Journals Publishing in Environmental Sciences and Public Health' has been provisionally accepted for publication in PLOS Global Public Health.

Best regards,

Zahra Zeinali, MD MPH

Academic Editor

Dear Kim, dear authors

Thank you for submitting this manuscript to PLoS GPH.

I am happy to confirm that your latest revisions have been well received and we can proceed with sending this submission for publication.

Please don't hesitate to reach out if you have any questions.

Kind regards,

Dr. Zahra Zeinali